# Effects of Soil Rhizobia Abundance on Interactions between a Vector, Pathogen, and Legume Plant Host

**DOI:** 10.3390/genes15030273

**Published:** 2024-02-22

**Authors:** Pooja Malhotra, Saumik Basu, Benjamin W. Lee, Liesl Oeller, David W. Crowder

**Affiliations:** Department of Entomology, Washington State University, Pullman, WA 99164, USA; pooja.malhotra@wsu.edu (P.M.); benjamin.w.lee@wsu.edu (B.W.L.); elisabeth.oeller@wsu.edu (L.O.); dcrowder@wsu.edu (D.W.C.)

**Keywords:** soil rhizobia, pea aphid, PEMV, defensive genes, symbiosis, abundance

## Abstract

Soil rhizobia promote nitrogen fixation in legume hosts, maximizing their tolerance to different biotic stressors, plant biomass, crop growth, and yield. While the presence of soil rhizobia is considered beneficial for plants, few studies have assessed whether variation in rhizobia abundance affects the tolerance of legumes to stressors. To address this, we assessed the effects of variable soil rhizobia inoculum concentrations on interactions between a legume host (*Pisum sativum*), a vector insect (*Acyrthosiphon pisum*), and a virus (*Pea enation mosaic virus*, PEMV). We showed that increased rhizobia abundance reduces the inhibitory effects of PEMV on the nodule formation and root growth in 2-week-old plants. However, these trends were reversed in 4-week-old plants. Rhizobia abundance did not affect shoot growth or virus prevalence in 2- or 4-week-old plants. Our results show that rhizobia abundance may indirectly affect legume tolerance to a virus, but effects varied based on plant age. To assess the mechanisms that mediated interactions between rhizobia, plants, aphids, and PEMV, we measured the relative expression of gene transcripts related to plant defense signaling. Rhizobia concentrations did not strongly affect the expression of defense genes associated with phytohormone signaling. Our study shows that an abundance of soil rhizobia may impact a plant’s ability to tolerate stressors such as vector-borne pathogens, as well as aid in developing sustainable pest and pathogen management systems for legume crops. More broadly, understanding how variable rhizobia concentrations can optimize legume-rhizobia symbiosis may enhance the productivity of legume crops.

## 1. Introduction

Legumes are a critical plant family that are often rotated with cereal and vegetable crops to provide nutrients to the soil and promote the sustainability of crop systems. Legume plants associate with rhizobial bacteria in the soil to regulate nutrient cycling, and these associations between plants and microbial symbionts largely drive global nutrient cycles [1]. Soil microbes supply plants with both nitrogen and phosphorus to maintain and boost a plant’s productivity [2] and, in exchange, soil microbes receive carbon and sugars from their hosts that aid in growth and metabolism [3]. While these interactions are considered mutualistic, an abundance of soil microbes can impact their ability to associate with plants and the benefits provided [4,5]. For example, if soil microbes are too abundant, they may consume too many nutrients from plants and become functional parasites [4].

Rhizobia are a group of soil bacteria that form symbiotic relationships with legumes, fixing 70 million metric tons a year of atmospheric nitrogen through their formation of nodules on roots [6]. Symbioses between legume plants and soil rhizobia can alter host physical and chemical defense responses, as well as promote the tolerance of the hosts to pathogens and herbivore pests [7,8]. However, different biotic and abiotic stressors can hamper symbioses between soil rhizobia and legume hosts through inhibiting root nodule formation, or by interfering with nitrogen fixation [9,10]. It is possible that symbioses between soil rhizobia and legume hosts, and the resulting nitrogen fixation, may depend on soil rhizobia abundance [11,12,13,14]. If there is a higher than optimal abundance of rhizobia, it may lead to an excessive consumption of nutrients to the detriment of host plants, but a low abundance of soil rhizobia may provide sub-optimal benefits to their hosts by limiting the rhizobia-induced benefits to the legume hosts [5]. The optimal concentration of rhizobia needed may depend on the soil conditions, competitiveness and effectiveness of rhizobia strains, and the overall levels of a plant’s encountered stress.

The optimization of soil rhizobia abundance could enhance the tolerance of legumes to biotic stressors (including both pathogens and pests) and increase their overall growth and yield [15,16]. Few studies have assessed the effects of soil rhizobia abundance regarding these factors in the presence of multiple stressors [4,17,18]. We predict that interactions among legume hosts and rhizobial bacteria may affect tolerance to biotic and abiotic stressors through different plant-mediated mechanisms. For example, rhizobia alter plant defense against herbivores or pathogens through regulating defense-related phytohormones, such as salicylic acid, jasmonic acid, abscisic acid, and ethylene [19]. However, due to the fact that plants produce many of these same phytohormones in response to abiotic stress [20,21], rhizobia may alter tolerance to many stressors simultaneously. We predict that an abundance of rhizobia could affect all these factors as they may alter plant traits that mediate tolerance to stress.

Nearly all studies testing the effects of rhizobia on plant health have used commercial strains with a single concentration, and they focused on a single stressor. In addition, we have limited data on how variation in rhizobia abundance may affect plant traits and stress tolerance. Here, we assessed multitrophic interactions between a legume host (pea, *P. sativum*), soil rhizobia (*Rhizobium leguminosarum biovar. viciae*), a herbivore vector (pea aphid, *A. pisum*), and a vector-borne virus (PEMV). Specifically, we experimentally assessed how varying abundances of soil rhizobia affected plant defense responses to the aphid vector and virus, and on PEMV transmission. Through a combination of manipulative experiments and gene expression studies, we show that the benefits of soil rhizobia for plant tolerance to biotic stressors depend on overall abundance of rhizobia in the soil, in ways that could impact the management of legume crops.

## 2. Materials and Methods

### 2.1. Study System

In the Pacific Northwest of the United States, legume hosts, pea aphid vectors, PEMV, and rhizobial bacteria all co-occur in natural and managed ecosystems [22]. In this region, *A. pisum* is the primary aphid vector of PEMV, which has outbreaks every six to nine years and reduces legume yield by 40% or more [23]. While rhizobia can aid in promoting legume tolerance to biotic and abiotic stressors [9,24] it is unknown whether varying rhizobia abundance affects the indirect benefits that plants receive in terms of increased tolerance to aphid vectors and PEMV.

Here, we assessed how varying concentrations of soil rhizobia inoculum affected interactions between pea plants, pea aphid vectors, and the vector-borne virus, PEMV. All aphids were reared in greenhouses (16:8 h light:dark; 22:17 °C light:dark) on *P. sativum* (cv. Banner). We reared both viruliferous and non-viruliferous *A. pisum* colonies that both originated from a field collection of viruliferous *A. pisum* that occurred in 2012. To create the non-viruliferous colony, 50 viruliferous *A. pisum* adults were put in petri dishes to reproduce for 3 days. All nymphs were non-viruliferous as PEMV is not maternally transmitted and, thereafter, non-viruliferous individuals were reared on uninfected plants; the viruliferous colony was reared on infected plants in a separate greenhouse [25].

### 2.2. Experimental Design and Data Collection

Our experiment was conducted in a greenhouse (16:8 h light:dark; 22:17 °C light:dark) with pea plants (cv. Banner) in potting soil (Sunshine^®^ LC1, Sun Gro Horticulture, Agawam, MA, USA). The experiment was a 2 × 2 × 8 factorial with eight rhizobia treatments: (i) no rhizobia and (ii–viii) seven rhizobia abundances (3.75, 7.5, 15, 30, 60, 120, 240 mg inoculum); two *A. pisum* treatments: (i) no *A. pisum* and (ii) viruliferous *A. pisum*; at two times: (i) 2- and (ii) 4-week-old plants. For treatments with rhizobia, we mixed N-Dure, a peat-based inoculant with *R. leguminosarum* biovar. *viciae*, with pea seeds based on the manufacturer’s protocol (Verdasian life Sciences, Cary, NC, USA). The seven abundance values were a geometric series around the median.

Following rhizobia treatments, plants were grown for 2- or 4-weeks at 75% soil moisture. After this time, ten 5-day old PEMV-infectious adults that fed for 24 h were placed on 2- and 4-week-old pea plants. We had eight replicates for each treatment, and two time blocks, for 256 total replicates. After feeding, *A. pisum* were removed using an aspirator, and their removal was ensured by observing plants for 72 h. All treatments were conducted on individual plants in mesh ‘bug dorms’ (0.6 × 0.6 × 0.6 m). Plants were allowed to develop virulence for five days after the treatments ended by carefully removing all aphids from plants. Tissue was then collected to measure the root and shoot fresh weight, PEMV titer, and defense gene expression. Nodule numbers from each plant were visually counted after washing the roots under tap water. Tissue samples from the aboveground portion of pea plants were collected, flash frozen in liquid nitrogen (N_2_), and snap chilled in dry ice before being stored in a −80 °C freezer.

To measure relative PEMV-1 and PEMV-2 accumulation, frozen samples were ground into a powder in liquid N_2_, using mortar and pestles. Homogenized tissue (50–100 mg) was used for total RNA extraction using Promega SV total RNA isolation kits (Promega, Madison, WI, USA), and cDNAs were synthesized from 1 µg of total RNA using Bio-Rad iScript cDNA synthesis kits. PEMV-1 and two specific primers (Table 1) were used in qRT-PCR reactions (10 µL) containing 3 µL of ddH_2_O, 5 µL of iTaq Univer SYBR Green Supermix, 1 µL of diluted primer mix (forward and reverse [concentration 10µM]), and 1 µL of diluted (1:25) cDNA template. qRT-PCR was performed using a program with an initial denaturation of 3 min at 95 °C, followed by 40 cycles of denaturation at 95 °C for 15 sec, annealing for 30 sec at 60 °C, and extension for 30 sec at 72 °C. For melting curve analysis, a dissociation step cycle was added (55 °C for 10 sec, and then 0.5 °C for 10 sec until 95 °C). The relative accumulation of PEMV-1 and -2 were then calculated using the delta-delta Ct method, 2^–∆∆Ct^) with *Psβ-tubulin* as a housekeeping gene [26].

To assess if phytohormones mediated interactions between pea plants, rhizobia, *A. pisum*, and PEMV, we measured the expression of four defense gene transcripts for 2- and 4-week-old plants. *Pathogenesis-related protein 1* (*PR1*) is associated with salicylic acid mediated defense and triggers systemic acquired resistance [27,28]. *Lipoxygenase 2* (*LOX2*) works upstream of jasmonic acid biosynthesis, a phytohormone hormone used in defense against herbivore pests [29,30]. Similarly, *1-aminocyclopropane-1-carboxylic acid synthases 2* (*ACS2*) is associated with ethylene synthesis [31,32], and *Aldehyde oxidase 3* (*AO3*) is associated with abscisic acid synthesis [33,34]. Besides ABA biosynthesis, AO3 is also involved in the production of reactive oxygen species (ROS) which, together with abscisic acid, provides resistance against pathogen infection [35]. All gene transcripts were chosen from literature on defense genes in peas, and sequences of these genes were obtained using the accession numbers of pea defense genes in NCBI, or by using Pea Marker Database [36]. Gene specific primers (Table 1) for Quantitative real time (qRT)-PCR of each defense gene tested were designed using the IDT Primer Quest Tool.

### 2.3. Data Analysis

Analyses were conducted in R v. 4.0.3 [37]. We used linear regression models to assess whether rhizobia abundance, aphid treatment, and their interaction affected nodule counts, shoot weight, and root weight using the “lme4” package; separate models were run for 2- or 4-week-old plants. Nodule counts were assessed through a Poisson distribution; shoot and root weight were assessed through Gaussian distributions. For PEMV and gene expression, we ran quadratic linear regression models on PEMV-1, PEMV-2, and each relative gene transcript value as responses, with rhizobia abundance, aphid treatment, and the interaction as fixed effects. Analyses for viral titer and gene expression were run on cycle threshold values (Ct), and 2^–∆∆Ct^ (relative expression) was calculated with parameter estimates from the models; separate models were run for 2- and 4-week-old plants. Estimated marginal mean of Ct values and standard errors were generated using the “emmeans” package [38]. Control treatments without rhizobia and aphids were used as reference values to calculate expression fold change; all values for other treatments were represented as the relative change in gene expression relative to the control treatment (which had a value of 1 by default). For gene expression, we ran quadratic linear regression models on PEMV-1, PEMV-2, and each relative gene transcript value as responses, with rhizobia abundance, aphid treatment, and their interaction as fixed effects. Separate models were run for 2- and 4-week-old plants. Analyses for viral titer and gene expression were run on cycle threshold values (Ct), and 2^–∆∆Ct^ (relative expression) was calculated using parameter estimates from the model. Estimated marginal mean of Ct values and standard errors were generated using the “emmeans” package in R [38]. Control treatments without rhizobia and aphids were used as reference values to calculate expression fold change; all values for other treatments are thus represented as the relative change in gene expression relative to the control treatment (which had a value of 1 by default).

## 3. Results

### 3.1. Effects of Rhizobia Abundance and PEMV Infection on Plant Growth Traits

In 2-week-old plants, a greater rhizobia abundance increased nodule counts (Z = 2.74, *p* = 0.005), but in 4-week-old plants this trend was reversed (Z = −2.68, *p* = 0.007, Figure 1A). Similarly, increased rhizobia abundance increased root weight in 2-week-old plants (*t* = 2.47, *p* = 0.014), but lowered root weight in 4-week-old plants (*t* = 2.43, *p* = 0.015) (Figure 1B). Plants exposed to PEMV had fewer nodules than plants not exposed to aphids (2-week-old: Z = −6.98, *p* < 0.001; 4-week-old: Z = 5.15, *p* < 0.001), but there were no interactions between rhizobia abundance and PEMV on nodules (*p* > 0.10) (Figure 1A). PEMV infection also significantly reduced plant root weight for 2- (*t* = −2.70, *p* = 0.007) and 4-week-old (*t* = −2.27, *p* = 0.020, Figure 2B) plants. Like nodules, there was no significant interaction between rhizobia abundance and PEMV on root weight (*p* > 0.10) (Figure 1B).

Like root weight, PEMV reduced shoot weight regardless of rhizobia abundance (2-week-old plants: *t* = −2.32, *p* = 0.020; 4-week-old plants: *t* = −2.47, *p* = 0.014, Figure 1C). Rhizobia abundance alone did not affect pea shoot weight (*p* > 0.10), but there was a significant interaction between rhizobia abundance and PEMV (*t* = 2.38, *p* = 0.018, Figure 1C), where PEMV infection had a significant negative effect on shoot weight only when there were lower rhizobia abundances (Figure 1C). At higher levels of rhizobia abundance, there were no differences between PEMV-infected and uninfected plants (Figure 1C).

### 3.2. Effects of Rhizobia Abundance on PEMV Accumulation

Rhizobia concentration did not strongly affect the accumulation of the PEMV-1 virus in plants that were 2- (*t* = 0.18, *p* = 0.85) or 4-week-old (*t* = −0.78, *p* = 0.43, Figure 2A). Similarly, rhizobia concentration did not affect the accumulation of the PEMV-2 virus in 4-week-old plants (*t* = 0.40, *p* = 0.69), but there was a significant reduction in PEMV-2 with higher rhizobia concentrations in 2-week-old plants (*t* = −5.36, *p* < 0.001, Figure 2B).

### 3.3. Effects of Rhizobia Abundance and PEMV Infection on Defense Gene Expression in Peas

Increasing rhizobia abundance decreased relative gene expression of *ACS2* in 2-week-old plants (*t* = 1.93, *p* = 0.056), but this effect was reduced when PEMV was present (*t* = −1.99, *p* = 0.049) (Figure 3A). In contrast, increasing rhizobia abundance increased relative gene expression of *PR1* in 2-week-old plants (*t* = −2.136, *p* = 0.035), although there was no strong effect in 4-week-old plants (*p* > 0.10) (Figure 3D). In contrast, rhizobia abundance did not strongly affect relative gene expression of *AO3* or *LOX2* in 2- or 4-week-old plants (*p* > 0.10 for all analyses, Figure 3B).

PEMV infection had variable effects on defense genes. In 4-week-old plants, PEMV decreased *ACS2* levels (*t* = 2.61, *p* = 0.01), although this effect was weakest when there was lower rhizobia abundance (*t* = −2.44, *p* = 0.01) (Figure 3A). In contrast, PEMV infection increased *AO3* (*t* = −10.3, *p* = 0.001, Figure 3B) and *LOX2* (*t* = −4.08, *p* < 0.001, Figure 3C) levels in 2-week-old plants, and *PR1* levels in both 2- (*t* = −15.44, *p* < 0.001) and 4-week-old plants (*t* = −2.17, *p* = 0.03) (Figure 3D). However, effects of PEMV on *PR1* were reduced by increased rhizobia abundance (*t* = 2.94, *p* = 0.004). PEMV infection, or the interaction between PEMV and rhizobia abundance, did not affect other relative gene expression levels (*p* > 0.10).

## 4. Discussion

We investigated the effects of varying soil rhizobia abundance on an aboveground vector-borne persistent plant virus, *Pea enation mosaic virus* (PEMV). Our study showed that soil rhizobia increased plant tolerance against PEMV, although effects were muted when plants were grown at relatively lower rhizobia abundances. Pea plants grown with rhizobia had increased root nodule numbers and root weight. High abundance of soil rhizobia also mediated the negative effects of PEMV infection on root nodule counts and shoot weight. Soil rhizobia increased the expression of two genes related to the defense hormones ethylene and abscisic acid, but not genes affecting jasmonic or salicylic acid. Along with our prior studies, we showed that mutualistic soil rhizobial bacteria play a key role in mediating plant tolerance to biotic stressors (including both vector-borne viruses and non-vector pests) by altering plant defense chemistry and nutritional quality [9,24]. However, the effect of rhizobial bacteria on plant tolerance in the presence of an aphid vector and a vector-borne pathogen was also shown to be mediated by rhizobia abundance, demonstrating the complexity of plant-insect-pathogen-microbe interactions in agroecosystems.

Our results are consistent with studies that show the benefits of soil rhizobia on plant growth and productivity, and the reduced susceptibility to herbivores, pathogens, and drought [39,40,41,42,43,44,45]. For example, legume plants grown with soil rhizobia have greater root weight and shoot weight, and reduced cellular, oxidative, and toxicity-related stress responses than plants grown without rhizobia [46,47,48,49,50]. Conversely, herbivores and pathogens can suppress legume-rhizobia symbiosis by suppressing nodule formation (e.g., by reducing nodule sizes, nodule numbers, nodule weight, dry biomass, and by reducing expression of genes involved in different steps of nodule development), and ultimately decreasing nitrogen fixation [9,24,51,52]. Our prior study also showed that interactions between soil microbes and their hosts, and indirect effects on herbivore insects and pathogens, depend on the soil conditions and abiotic factors (Malhotra et al. unpublished). Here, we show complex multitrophic ecosystem interactions among plants, insect vectors, plant pathogens, and soil microbes also depend on the abundance of soil microbes.

Aboveground vector-borne pathogens have been found to inhibit plant mutualistic soil microbes by interfering with nitrogen fixation [51]. For example, *Southern bean mosaic virus*, *cucumber mosaic virus*, *alfalfa mosaic virus*, *tobacco ringspot virus*, and *bean yellow mosaic virus* were reported to reduce nodulation formation in different legume hosts, ultimately leading to compromised N2 fixation and legume growth [10,53,54,55]. Therefore, optimizing legume-rhizobia symbiosis by varying the rhizobia abundance, or by using an elite rhizobia strain with better legume accessions, can maximize nitrogen fixation, as well as maximize rhizobia-induced benefits in legumes [56,57]. An increased biological nitrogen fixation due to the improved legume rhizobia symbiosis can stimulate and maximize legume growth, development and activation of defense response, and achieve enhanced protection against aboveground pathogens and insects [47,58]. Therefore, optimizing legume rhizobia symbiosis is crucial for developing sustainable pest and pathogen management systems, and improving legume cropping systems to meet the enhanced global agri-food demand associated with rapidly increasing global population.

Our study provides further evidence that varying rhizobia abundance may change legume rhizobia symbiosis through altering nodule formation, root and shoot growth, prevalence of a vector-borne plant virus (PEMV), and insight into plant defense responses through varying symbiotic interactions with legume hosts. We showed that the increased rhizobia abundance reduced inhibitory effects of PEMV on nodule formation and root growth in 2-week-old plants. However, these trends were reversed in 4-week-old plants. Rhizobia abundance did not affect shoot growth or virus prevalence. These results indicate that higher rhizobia concentrations initially promoted nodule formations and shoot growth, but these effects diminished over time. Higher concentrations of rhizobia had no effects on limiting the negative effects of PEMV on root growth, but lower concentrations did. Similarly, higher rhizobia concentrations had no effect on PEMV-prevalence, and negatively affected expression of defense genes related to phytohormone signaling. Thus, higher concentrations of soil rhizobia do not necessarily improve all aspects of plant-microbe symbiosis through providing rhizobia-induced benefits to legume hosts. Rhizobia supply ammonia or amino acids through biological nitrogen fixation (cost), and return organic acids as carbon source and energy (benefit). In nature, this legume-rhizobia resource mutualism maintains a subtle cost to benefit ratio [59]. Higher abundances of rhizobia may become parasitic, taking a high proportion of nutrients from their host plants, and may not be able to mitigate negative effects of biotic stressors (e.g., a vector-borne pathogen, PEMV in this case). This is consistent with our results showing higher rhizobia abundance led to decreased stress tolerance and induction of phytohormone-mediated defense responses. Similarly, common milkweed, *Asclepias syriaca*, was tested for traits associated with plant growth and defense with varying concentrations of another plant growth promoting soil microbe, arbuscular mycorrhizal fungi; the study showed that the growth and defense expression still depends on the optimal mutualism [4]. Determining the optimal concentration of non-pathogenic soil microbiota mediating maximum symbiotic association with legume root is crucial for obtaining ideal growth and defense expression.

We found that soil rhizobia mediate complex interactions between peas, pea aphid vectors, and the vector-borne pathogen, PEMV through affecting growth and altering plant tolerance to vector-borne pathogens. Defense related phytohormones include salicylic acid (SA), jasmonic acid (JA), abscicic acid (ABA), and ethylene (ET). Salicylic acid activates anti-pathogens defense responses, and jasmonic acid and ethylene are common anti-herbivores defense hormones, while abscisic acid promotes stress tolerance against both biotic and abiotic stressors via diverse mechanisms. The interaction among different defense-related phytohormone signaling pathways is crucial in coordinating plant growth, resource allocation, and defense against subsequent attackers through synergistic and antagonistic interactions, a phenomenon called “phytohormone cross-talk” [60,61]. We found that rhizobia presence significantly increased expressions of ethylene-biosynthetic gene, *ACS2,* and abscisic acid biosynthetic gene, *AO3*, while having no significant effect on *LOX2* (jasmonic acid-biosynthetic gene) and *PR1* (salicylic acid-responsive gene) (Table 1). PEMV infection negatively regulated the effects of soil rhizobia on relative expression of all four genes. Interestingly, higher concentrations of soil rhizobia were found to have inhibitory effects on *ACS2* expression, and this failed to show any significant effect on the other 3 genes (*PR1, LOX2*, and *AO3*, respectively; Table 1). However, plant growth stages (2- and 4-week-old) had some effects on defense gene expression following complex interactions with pea aphid vectors, PEMV and varying rhizobia abundance.

Although the evolutionary trajectories and genetic drivers of legume-rhizobia symbiosis have been tested in many studies, they are still poorly understood [5,59,62]. Few studies have investigated the effects of optimization and competitiveness of legume-rhizobia symbiosis and the mechanistic details of how these interactions interfere with legume tolerance to various stress responses [57,63]. For the first time, we show the effects of rhizobia abundance with respect to tri-trophic ecosystem involving aboveground vector-virus pathosystem (a vector, i.e., pea aphid and a vector-borne virus, i.e., PEMV) and legume hosts [9,64,65]. Our study could aid in progressing sustainable agriculture by improving crop health, productivity, and by achieving enhanced protection against aboveground biotic stressors by inducing host defensive responses.

## 5. Conclusions

The human population continues to grow, creating a need to continue to increase our global food supply and therefore the efficiency of production by improving growth and yield with fewer inputs. Unfortunately, the total amount of natural resources that supplies nitrogen fertilizer to crops are decreasing. Because of this, greater emphasis is being placed on alternatives to nitrogen fertilizer, such as improved nitrogen fixation. This process could be achieved by selecting elite strains of rhizobia and more suitable legume varieties in agroecosystems. An investigation of rhizobia abundance, competitiveness, and symbiosis with varying legume varieties is critical to the optimization of legume-rhizobia symbiosis, and assessment of the competitiveness of rhizobia with the legume variety and their ability to fix biological nitrogen in legume needs careful investigation. As legumes are commonly rotated with cereal grains (corn, rice, wheat, barley etc.) in agroecosystems around the world, understanding how the abundance of soil microbes, could lead to more effective management of biological nitrogen fixation and improved productivity of many crops.

Our research study is among the first to assess how varying abundances of plant-growth promoting soil rhizobia affected plant responses to an aboveground vector-virus pathosystem by altering plant growth traits, virus prevalence, and plant defenses. We clearly show that the benefits of soil rhizobia-induced plant tolerance to biotic stressors depend on the overall abundance of rhizobia in the soil. This suggests management of soil rhizobia abundance may be important for the management of legume crops. Seeking to use optimal concentrations of compatible soil rhizobia inoculum, which may differ for legume accessions and based on variable environmental conditions, may provide an effective and novel management approach to manage devastating herbivores and plant pathogens they transmit. In addition to using an optimal abundance of compatible and superior strains of soil rhizobia inoculum, selection of better legume accessions that are more able to access benefits from rhizobia through the maintenance of symbiotic relationships may also help optimizing the legume rhizobia symbiosis and nitrogen fixation. The ability to modify legume-rhizobia symbiosis by selectively picking up a desired abundance of superior and/or compatible soil rhizobia, as well as a better legume accession, may aid in sustainable pest and pathogen management more effectively, and lead to an enhancement of productivity for the future to meet the increasing global agri-food demands.

## Figures and Tables

**Figure 1 genes-15-00273-f001:**
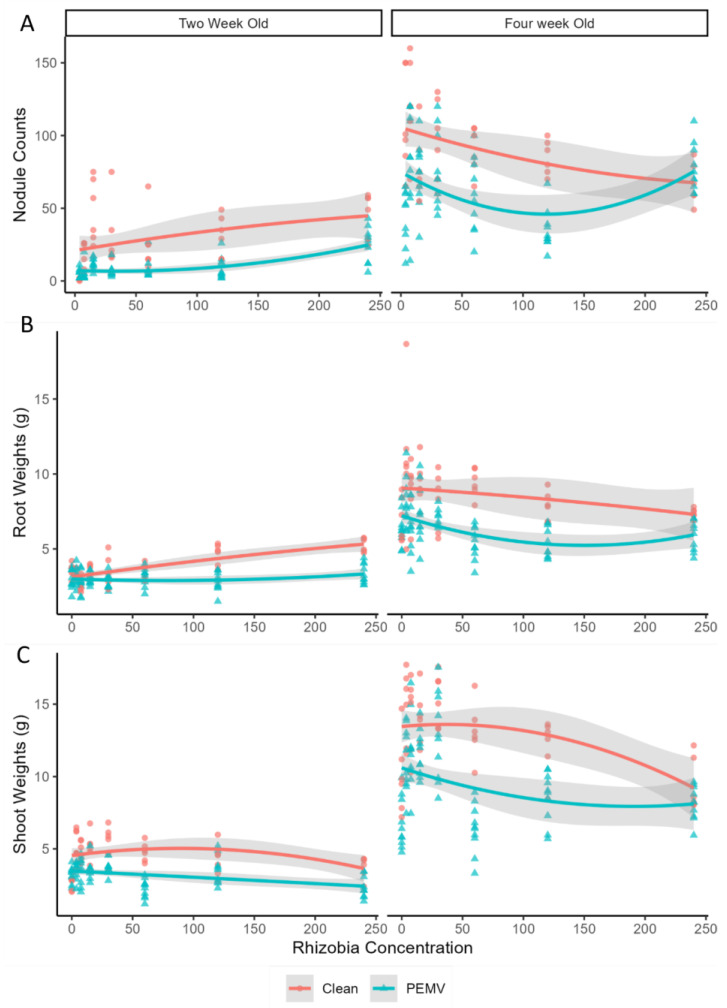
Effects of rhizobia abundance (mg/100 µL) and PEMV infection on pea (**A**) nodule numbers, (**B**) fresh root weights, and (**C**) fresh shoot weights. Red circles reflect values from individual biological replicates with no infectious aphids (*n* = 8 per treatment; 128 total per panel), and red lines indicate the fit of the quadratic regression models. Cyan triangles reflect values from individual biological replicates with infectious aphids (*n* = 8 per treatment; 128 total per panel), and cyan lines reflect the fit of the quadratic regression models. In each panel, the 95% confidence intervals around the quadratic regression models are shown in grey bands.

**Figure 2 genes-15-00273-f002:**
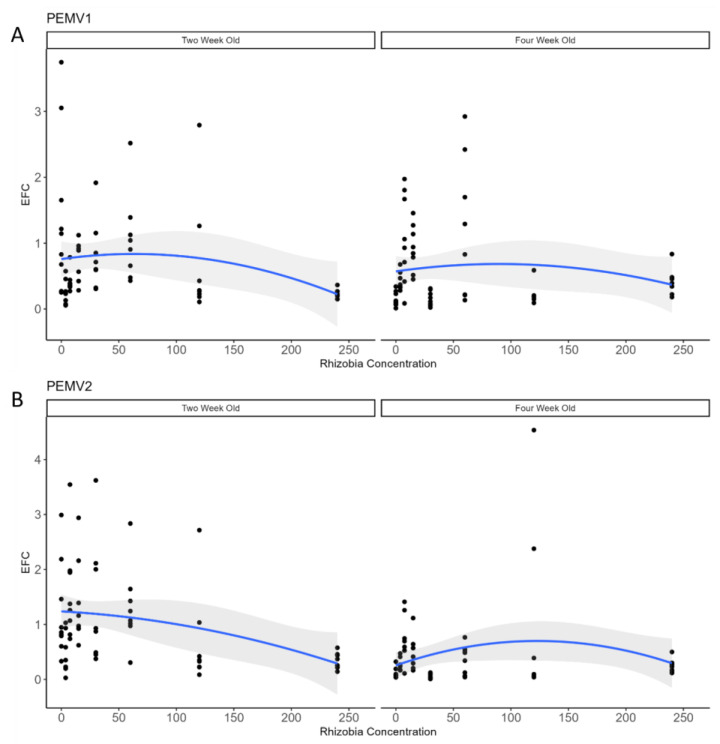
Effects on rhizobia abundance (mg/100 µL) on PEMV-1 and PEMV-2 accumulation in 2 and 4-week-old peas (measured as effective fold change; EFC). Individual dots represent various biological replicates within each treatment (*n* = 8 replicates per treatment; 56 total per panel). In each panel, the fit of a quadratic regression model is shown as a blue line, and the 95% confidence intervals around the models are shown in grey.

**Figure 3 genes-15-00273-f003:**
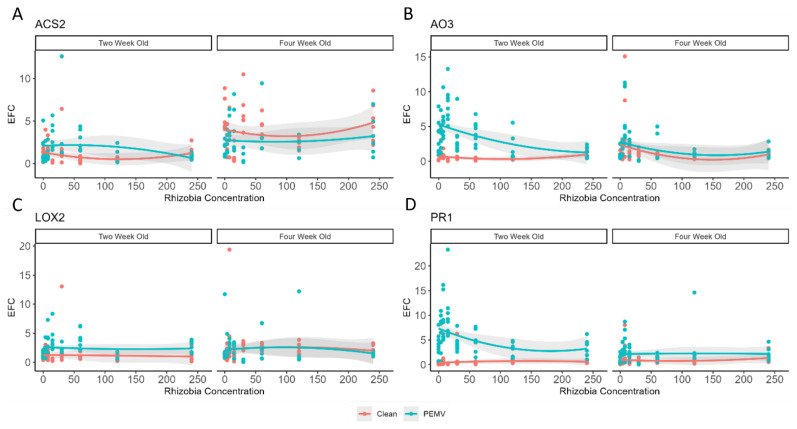
Relative transcript accumulation of pea defense genes associated with phytohormones based on the rhizobia abundance (measured as effective fold change; EFC): (**A**) *ACS2,* (**B**) *AO3*, (**C**) *LOX2*, and (**D**) *PR1*. Red circles reflect values from individual biological replicates with no infectious aphids (*n* = 8 per treatment; 128 total per panel), and red lines indicate the fit of the quadratic regression models. Cyan circles reflect values from individual biological replicates with infectious aphids (*n* = 8 per treatment; 128 total per panel), and cyan lines reflect the fit of the quadratic regression models. In each panel, the 95% confidence intervals around the quadratic regression models are shown in grey bands.

**Table 1 genes-15-00273-t001:** List of primers used for this study.

Gene	Primer Sequences	NCBI Accession No.	Amplicon Size (bp)
*PEMV-1 FP* *PEMV-1 RP*	GCAATCCTACAGGACCTTCATA CTCATCGTCTTCCGTGTCATC	HM439775.1	121 bp
*PEMV-2 FP* *PEMV-2 RP*	TGCTAGGAGAGGTGGAGATATG GCAATTGAGTAGGGTGGGTAAA	JF713436.1	130 bp
*PsPR1 F* *PsPR1 R*	TGGGGCAGTGGTGACATAAC TGCGCCAAACAACCTGAGTA	LT635896	178
*Lox2 F* *Lox2 R*	GCAACCAAGTGACGAAGTCTA GGAGACCCGATTGTAAGGTATTT	PsCam 059875	95
*PsAO3 F* *PsAO3 R*	TTATAGGACACAGGCTAGCTCAGCA TGACACAAGCTTATTCAGCATGACA	EF491600.1	127
*PsACS2 F* *PsACS2 R*	GGCATAGTAATTTGAGGTTGAGCC GCCCCAACATTTAAAGGACCTATTA	AF016459	103
*β-tubulin* FP *β-tubulin* RP	GTAACCCAAGCTTTGGTGATC ACTGAGAGTCCTGTACTGCT	X54844.1	203

## Data Availability

The data that support the findings of this study are available on request from the corresponding author.

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
