# Peer review of "Effects of Soil Rhizobia Abundance on Interactions between a Vector, Pathogen, and Legume Plant Host"

_genes, 2024, doi:10.3390/genes15030273_

Round 1
Reviewer 1 Report
Comments and Suggestions for Authors
The paper is original and well focused on a specific problem, as it is the interaction about symbiotic microorganisms, pathogen and leguminous plants.
The results are interesting and provide novel information which may be of interest for the readers of "Genes". Although there are some minor points which should be adressed in order to increase the clarity of the paper.
Figure 1: Indicate what does the gray zone represent and how it was calculated. What does each circle and triangle represents? which is the n number for each point? Technical or biological replicates?
Figure 3: Indicate what does the gray zone represent and how it was calculated. Which is the n number for each point?
Conclusion: is too generic and quite speculative, arguing what could be done to improve agroecosystems. Seems more like an introduction. Please, make a brief description of the main results of the present work.
Author Response
The paper is original and well focused on a specific problem, as it is the interaction about symbiotic microorganisms, pathogen and leguminous plants.
The results are interesting and provide novel information which may be of interest for the readers of "Genes". Although there are some minor points which should be adressed in order to increase the clarity of the paper.
We thank the reviewer for their feedback on our manuscript and suggestions for improvement
Figure 1: Indicate what does the gray zone represent and how it was calculated. What does each circle and triangle represents? which is the n number for each point? Technical or biological replicates?
We agree the legend needed to be clearer, and we updated the caption as follows (L192-198):
“Figure 1. Effects of rhizobia abundance (mg/100 µL) and PEMV infection on pea (A) nodule numbers, (B) fresh root weights, and (C) fresh shoot weights. Red circles reflect values from individual biological replicates with no infectious aphids (n = 8 per treatment; 128 total per panel), and red lines indicate the fit of the quadratic regression models. Green triangles reflect values from individual biological replicates with infectious aphids (n = 8 per treatment; 128 total per panel and green lines reflect the fit of the quadratic regression models. In each panel, the 95% confidence intervals around the quadratic regression models are shown in grey bands.”
A similar change was made to Figure 2 (L206-210):
“Figure 2. Effects on rhizobia abundance (mg/100 µL) on PEMV-1 and PEMV-2 accumulation in 2 and 4-week old peas (measured as effective fold change; EFC). Individual dots represent various biological replicates within each treatment (n = 8 replicates per treatment; 56 total per panel). In each panel, the fit of a quadratic regression model is shown as a blue line, and the 95% confidence intervals around the model are shown in grey.”
Figure 3: Indicate what does the gray zone represent and how it was calculated. Which is the n number for each point?
We agree the legend needed to be clearer, and we updated the caption as follows (L226-235):
“Figure 3. Relative transcript accumulation of pea defense genes associated with phytohormones based on the rhizobia abundance (measured as effective fold change; EFC): (A) ACS2, (B) AO3, (C) LOX2, and (D) PR1. Red circles reflect values from individual biological replicates with no infectious aphids (n = 8 per treatment; 128 total per panel), and red lines indicate the fit of the quadratic regression models. Green triangles reflect values from individual biological replicates with infectious aphids (n = 8 per treatment; 128 total per panel), and green lines reflect the fit of the quadratic regression models. In each panel, the 95% confidence intervals around the quadratic regression models are shown in grey bands.”
Conclusion: is too generic and quite speculative, arguing what could be done to improve agroecosystems. Seems more like an introduction. Please, make a brief description of the main results of the present work.
We thank the reviewer for their comments, and we added a brief description of the main results in addition to the future direction of the of this present study in the conclusion section as suggested by the reviewer. The revised text on L349-365 is as follows:
“Our research study is among the first to assess how varying abundances of plant-growth promoting soil rhizobia affected plant responses to an aboveground vector-virus pathosystem by altering plant growth traits, virus prevalence, and plant defenses. We clearly show that the benefits of soil rhizobia induced plant tolerance to biotic stressors depended on the overall abundance of rhizobia in the soil. This suggest management of soil rhizobia abundance may be important for the management of legume crops. Seeking to use optimal concentrations of compatible soil rhizobia inoculum, which may differ for legume accessions and based on variable environmental conditions, may provide an effective and novel management approach to manage devastating herbivores and plant pathogens they transmit. In addition to using an optimal abundance of compatible and superior strains of soil rhizobia inoculum, selection of better legume accessions that are more able to access benefits from rhizobia through the maintenance of symbiotic relationships may also help optimizing the legume rhizobia symbiosis and nitrogen fixation. The ability to modify legume-rhizobia symbiosis by selectively picking up a desired abundance of superior and/or compatible soil rhizobia and a better legume accession may aid in sustainable pest and pathogen management more effectively and lead to enhance productivity for future to meet the increasing global agri-food demands.”

Reviewer 2 Report
Comments and Suggestions for Authors
The manuscript explores the intricate relationships between rhizobia abundance, pea plants, and the vector-borne pea enation mosaic virus (PEMV). The study investigates the effects of varying rhizobia abundance on plant growth, tolerance to PEMV, and defense gene expression. Overall, the manuscript addresses an important aspect of plant-microbe interactions and their influence on plant responses to viral infections.
- The manuscript would benefit from minor improvements in language flow and sentence structure. Some sentences are complex, and simplifying them would enhance overall readability.
- Ensure consistency in terminology throughout the manuscript. For instance, there is variation in terms like "2-week" and "2-week-old." Maintaining consistency will improve clarity.
- The abstract could be more concise and highlight the key findings, objectives, and implications of the study. Presenting a clear summary will attract readers and provide a quick overview of the manuscript's significance.
- The discussion section could be expanded to further elaborate on the implications of the findings, potential applications in agriculture, and suggestions for future research directions.
- The conclusion section could be strengthened by summarizing the main findings and reiterating their significance in the context of the broader field of plant-pathogen interactions.
- The manuscript presents a valuable contribution to the understanding of rhizobia-mediated effects on plant-virus interactions. The MS needs minor revisions to improve clarity and consistency, along with enhancements to the abstract, discussion, and conclusion sections. Overall, I appreciate the manuscript's objective and find it quite intriguing. Personally, I am working on a similar study on a larger scale, and I commend the authors for their excellent work!
Author Response
Reviewer 2:
The manuscript explores the intricate relationships between rhizobia abundance, pea plants, and the vector-borne pea enation mosaic virus (PEMV). The study investigates the effects of varying rhizobia abundance on plant growth, tolerance to PEMV, and defense gene expression. Overall, the manuscript addresses an important aspect of plant-microbe interactions and their influence on plant responses to viral infections.
We thank the anonymous reviewer for their feedback on our manuscript.
The manuscript would benefit from minor improvements in language flow and sentence structure. Some sentences are complex, and simplifying them would enhance overall readability.
We thank the reviewer for these suggestions. We have carefully edited the manuscript for readability and tried to simplify sentence structure throughout.
Ensure consistency in terminology throughout the manuscript. For instance, there is variation in terms like "2-week" and "2-week-old." Maintaining consistency will improve clarity.
Changed as suggested throughout the manuscript, all terminology is now consistent
The abstract could be more concise and highlight the key findings, objectives, and implications of the study. Presenting a clear summary will attract readers and provide a quick overview of the manuscript's significance.
We have improved the abstract by presenting the novelty of our study in the field of sustainable agriculture by efficiently managing devastating pests and pathogens in the revised manuscript. The revised abstract is as follows (L15-31):
“Soil rhizobia promote nitrogen fixation in legume hosts, maximizing tolerance to different biotic stressors, plant biomass, crop growth, and yield. While the presence of soil rhizobia is considered beneficial to plants, few studies have assessed whether variation in rhizobia abundance affects the tolerance of legumes to stressors. To address this, we assessed effects of variable soil rhizobia inoculum concentrations on interactions between a legume host (Pisum sativum), a vector insect (Acyrthosiphon pisum), and a virus (Pea enation mosaic virus, PEMV). We show that increased rhizobia abundance reduced inhibitory effects of PEMV on nodule formation and root growth in 2-week-old plants. However, these trends were reversed in 4-week-old plants. Rhizobia abundance did not affect shoot growth or virus prevalence in 2- or 4-week-old plants. Our results show rhizobia abundance may indirectly affect legume tolerance to a virus, but effects varied based on plant age. To assess mechanisms that mediated interactions between rhizobia, plants, aphids, and PEMV, we measured relative expression of gene transcripts related to plant defense signaling. Rhizobia concentrations did not strongly affect expression of defense genes associated with phytohormone signaling. Our study shows that abundance of soil rhizobia may impact the ability of plants to tolerate stressors such as vector-borne pathogens and aid in developing sustainable pest and pathogen management systems for legume crops. More broadly, understanding how variable rhizobia concentrations can optimize legume-rhizobia symbiosis may broadly enhance productivity of legume crops.”
The discussion section could be expanded to further elaborate on the implications of the findings, potential applications in agriculture, and suggestions for future research directions.
We have elaborated the discussion to address this feedback. In particular, we discuss potential applications in agriculture in specific places (L268-277):
“Therefore, optimizing legume-rhizobia symbiosis by varying rhizobia abundance or by using an elite rhizobia strain with better legume accessions can maximize nitrogen fixation and can maximize rhizobia induced benefits in legumes [56,57]. Increased biological nitrogen fixation due to improved legume rhizobia symbiosis can stimulate and maximize legume growth, development and activation of defense response and achieve enhanced protection against aboveground pathogens and insects [47,58]. Therefore, optimizing legume rhizobia symbiosis is crucial for developing sustainable pest and pathogen management systems and improving legume cropping systems to meet the enhanced global agri-food demand associated with rapidly increasing global population.”
The conclusion section could be strengthened by summarizing the main findings and reiterating their significance in the context of the broader field of plant-pathogen interactions.
This was a similar comment to Reviewer 2, and as mentioned above we completely rewrote the conclusion to summarize the main findings and their broader implications.
The manuscript presents a valuable contribution to the understanding of rhizobia-mediated effects on plant-virus interactions. The MS needs minor revisions to improve clarity and consistency, along with enhancements to the abstract, discussion, and conclusion sections. Overall, I appreciate the manuscript's objective and find it quite intriguing. Personally, I am working on a similar study on a larger scale, and I commend the authors for their excellent work!
We thank the reviewer for this positive feedback and look forward to seeing their future work.
